# Genetic and Clinical Profile of Retinopathies Due to Disease-Causing Variants in Leber Congenital Amaurosis (LCA)-Associated Genes in a Large German Cohort

**DOI:** 10.3390/ijms24108915

**Published:** 2023-05-17

**Authors:** Ditta Zobor, Britta Brühwiler, Eberhart Zrenner, Nicole Weisschuh, Susanne Kohl

**Affiliations:** 1Institute for Ophthalmic Research, Centre for Ophthalmology, University of Tübingen, Elfriede-Aulhorn Strasse 7, 72076 Tübingen, Germanysusanne.kohl@uni-tuebingen.de (S.K.); 2Department of Ophthalmology, Semmelweis University, 1085 Budapest, Hungary; 3Werner Reichardt Center for Integrative Neuroscience, University of Tübingen, 72076 Tübingen, Germany

**Keywords:** nherited retinal dystrophies, Leber congenital amaurosis (LCA), retinitis pigmentosa (RP), LCA-associated genes, genotype-phenotype correlation

## Abstract

To report the spectrum of Leber congenital amaurosis (LCA) associated genes in a large German cohort and to delineate their associated phenotype. Local databases were screened for patients with a clinical diagnosis of LCA and for patients with disease-causing variants in known LCA-associated genes independent of their clinical diagnosis. Patients with a mere clinical diagnosis were invited for genetic testing. Genomic DNA was either analyzed in a diagnostic-genetic or research setup using various capture panels for syndromic and non-syndromic IRD (inherited retinal dystrophy) genes. Clinical data was obtained mainly retrospectively. Patients with genetic and phenotypic information were eventually included. Descriptive statistical data analysis was performed. A total of 105 patients (53 female, 52 male, age 3–76 years at the time of data collection) with disease-causing variants in 16 LCA-associated genes were included. The genetic spectrum displayed variants in the following genes: *CEP290* (21%), *CRB1* (21%), *RPE65* (14%), *RDH12* (13%), *AIPL1* (6%), *TULP1* (6%), and *IQCB1* (5%), and few cases harbored pathogenic variants in *LRAT*, *CABP4*, *NMNAT1*, *RPGRIP1*, *SPATA7*, *CRX*, *IFT140*, *LCA5*, and *RD3* (altogether accounting for 14%). The most common clinical diagnosis was LCA (53%, 56/105) followed by retinitis pigmentosa (RP, 40%, 42/105), but also other IRDs were seen (cone-rod dystrophy, 5%; congenital stationary night blindness, 2%). Among LCA patients, 50% were caused by variants in *CEP290* (29%) and *RPE65* (21%), whereas variants in other genes were much less frequent (*CRB1* 11%, *AIPL1* 11%, *IQCB1* 9%, and *RDH12* 7%, and sporadically *LRAT*, *NMNAT1*, *CRX*, *RD3*, and *RPGRIP1*). In general, the patients showed a severe phenotype hallmarked by severely reduced visual acuity, concentric narrowing of the visual field, and extinguished electroretinograms. However, there were also exceptional cases with best corrected visual acuity as high as 0.8 (Snellen), well-preserved visual fields, and preserved photoreceptors in spectral domain optical coherence tomography. Phenotypic variability was seen between and within genetic subgroups. The study we are presenting pertains to a considerable LCA group, furnishing valuable comprehension of the genetic and phenotypic spectrum. This knowledge holds significance for impending gene therapeutic trials. In this German cohort, *CEP290* and *CRB1* are the most frequently mutated genes. However, LCA is genetically highly heterogeneous and exhibits clinical variability, showing overlap with other IRDs. For any therapeutic gene intervention, the disease-causing genotype is the primary criterion for treatment access, but the clinical diagnosis, state of the retina, number of to be treated target cells, and the time point of treatment will be crucial.

## 1. Introduction

Leber congenital amaurosis (LCA, MIM #204000) refers to a heterogeneous group of severe early infantile retinal dystrophies with typically extinguished electroretinograms (ERGs). It was first described by Theodor Leber in 1869 [1], who observed a group of young patients with poor vision and nystagmus due to a severe, recessively inherited early infantile onset rod-cone dystrophy (“classical” LCA). Later, a separate group of milder disease phenotypes, the so-called “early-onset severe retinal dystrophy” (EOSRD) or “severe early childhood onset retinal dystrophy” (SECORD), has been delineated [2]. LCA and EOSRD/SECORD together are the most severe of the early onset forms of all inherited retinal diseases (IRDs). They affect 20% of blind children and account for 5% of all IRDs [2,3]. The worldwide prevalence is 1: 30,000 to 1:80,000; in Germany, the estimated number of cases is ~2000 (source: Pro Retina Deutschland e. V.) [4].

With the advances in molecular diagnostic genetic technology, knowledge about the genetic background of LCA has expanded widely in recent years. Disease-causing variants associated with LCA have been identified in more than 25 genes, and different path mechanisms have been implicated, all of which result in a diverse range of retinal dysfunction and affected pathways, including the phototransduction, the visual cycle, and photoreceptor development/integrity [2,5]. Hence, the clinical phenotypes also exhibit extensive heterogeneity, including the onset and course of vision loss, involvement of the macular area, alteration in retinal structure, and residual function of the diseased photoreceptors [3,6,7]. The clinical diagnosis can be complicated by three aspects: (1) lack of a precise definition of LCA [1,8,9], (2) phenotypic overlap with other early onset IRDs [10,11], and (3) molecular genetic overlap, meaning one LCA-associated gene may cause various phenotypic forms of IRD [12]. Therefore, the ontology of early onset retinal dystrophies—and IRDs in general—has changed in recent years [13], and many diseases are now described and diagnosed according to their genotype (i.e., *ABCA4*-, *KCNV2*-, *RPE65*-associated retinopathy, bestrophinopathies—IRDs associated with disease-causing variants in *BEST1*) [14,15,16,17,18].

So far, at least 25 genes have been associated with LCA [12], and the genetic spectrum varies by geographic region and ethnic group. Mutations in *CEP290* (Gene ID: 80184; OMIM 610142), for instance are known to be frequent in Northern Europe [19,20,21,22]. This is due to a common deep intronic founder mutation, namely c.2991+1655A>G, which results in missplicing and a premature termination codon [19]. Due to the lack of this variant in other populations, it is remarkably less prevalent in Southern Europe [23,24] and Asia [25,26,27,28]. A recent study showed that the frequency in Denmark is also low [29], indicating that even within a predominantly Caucasian population, prevalence rates may not be generalized. In rare diseases, such as LCA, because of the limited number of patients, genetic frequencies can hardly be estimated and can even differ from country to country. Therefore, comprehensive national studies are important to obtain valid prevalence data.

As of today, the structural and functional associations seen in the various LCA/EOSRD genotypes are well described and understood [2,3,30]. The development and characterization of LCA/EOSRD animal models have shed light on the underlying pathogenesis and allowed the demonstration of successful rescue with gene replacement therapy and pharmacological intervention in multiple models [2,30]. These advancements have led to multiple completed, ongoing, and anticipated phase I/II and phase III gene therapy and pharmacological human clinical trials [2]. A promising first gene therapy has become available as “Luxturna” (voretigen neparvovec), a treatment for biallelic *RPE65*-mediated IRDs, was approved in December 2017 in the USA [31] and in November 2018 in Europe [32]. Therefore, genetic testing, clinical characterization of patients, and natural history studies remain essential to identify suitable patients and new targets. In addition, epidemiologic data are urgently needed to estimate the number of patients eligible for therapy in different geographic areas. In Germany, however, the genetic spectrum of this rare disease group has been poorly described, and a detailed phenotype-genotype correlation in a representative cohort is lacking [33].

The purpose of the current study is to describe the prevalence of disease-causing variants in LCA-associated genes in a large German cohort at a single clinical site and to give detailed clinical information on subgroups, aiming to pave the way for future gene therapies.

## 2. Results

### 2.1. Genetic Findings

A total of 105 patients (53 male, 52 female, age 3–76 years, mean 30.1 years at the time of data collection) from 87 families with confirmed disease-causing variants in LCA-associated genes were included. Altogether, disease-causing variants in 16 LCA-associated genes were found in our cohort (*AIPL1*, *CABP4*, *CEP290*, *CRB1*, *CRX*, *IFT140*, *IQCB1*, *LCA5*, *LRAT*, *NMNAT1*, *RD3*, *RDH12*, *RPE65*, *RPGRIP1*, *SPATA7*, and *TULP1*) (Table 1). Among these, the largest genetic groups were cases carrying biallelic variants in *CEP290* (22/105), *CRB1* (22/105), and *RPE65* (15/105) (Figure 1). Segregation analysis to confirm biallelism was available for 43 cases (41% of cases; 28 families, 32%). A list of the eight most frequent alleles in our cohort is shown in Table 2. A list of the genotypes of all patients and the established clinical diagnosis is shown in Appendix A.

In most cases, disease onset was reported in infancy or early childhood, but in a few cases first symptoms were noticed in middle aged children or in young adults. The distribution of disease onset (shown in Figure 2) also varied between genetic subgroups and consequently defined the clinical categorization. The most common clinical diagnosis was LCA (53%, 56/105) followed by RP (40%, 42/105), but also other forms of IRDs were seen (i.e., cone-rod dystrophy (CRD): 5%, 5/105, and congenital stationary night blindness (CSNB): 2%, 2/105) (Table 3).

Among the patients with the clinical diagnosis of LCA, ~50% of cases were caused by variants in *CEP290* (28%; 15/56) and *RPE65* (23%; 12/56), while variants in other genes were much less frequent (*CRB1* 11%, *AIPL1* 11%, *IQCB1* 7.5%, and *RDH12* 7.5%, and sporadically *LRAT, NMNAT1, CRX, RD3*, and *RPGRIP1*). If the initial clinical diagnosis was RP or SECORD/EOSRD, the compilation of responsible genes was different: *CRB1* (36%, 15/42) was the most common, followed by *RDH12* (24%, 10/42) and *CEP290* (16%, 7/42), while other genes were less frequent (Figure 3).

In total, 91 unique variants were identified. Among these, missense variants were most frequent (n = 44), followed by nonsense (n = 23), frameshift (n = 16), splicing variants (n = 7), and single exon deletion (n = 1).

### 2.2. Clinical Findings

In general, disease-causing variants in LCA-associated genes led to a severe phenotype (Table 1). However, there were exceptional cases with best corrected visual acuity (BCVA) up to 0.8 (Snellen), good maintained visual fields, and preserved photoreceptors in spectral domain optical coherence tomography (SD-OCT). Associated findings like nystagmus, night blindness, color vision disturbances, photophobia, cataract, and strabismus were frequent but non-specific (Table 4). The BCVA was severely reduced (0.12 ± 0.20; median (m): 0.03 (Snellen); for OD), with 29% (31/105) having no light perception, or light perception and 15% (16/105) being able to perceive hand motion. The visual field was severely restricted or not detectable in most cases (M ± SD: 1403 ± 2702 deg^2^; m: 23 deg^2^; for OD): In 41% (33/80), visual field (VF) was not preserved and in another five patients, VF was constricted <5 degrees (for OD). Thirty-seven percent (28/105) had no or old VF records, so no field in deg^2^ could be measured. The electroretinograms (ERG) (feasible in 84/105 cases) was mostly non-reproducible (73%, 61/84). Patients with non-reproducible responses were most frequently diagnosed with LCA (62%, 38/61), followed by RP (33%, 20/61) and CRD (5%, 3/61). Residual ERG responses were present in 21% (23/84); here, the most frequent diagnosis was RP (61%, 4/23), followed in descending order by LCA (17%, 4/23), CSNB (9%, 2/23), and CRD (9%, 2/23).

Funduscopy revealed typical signs of IRDs (e.g., pale optic discs, constricted vessels, pigment epithelial atrophy, pigment deposits, altered reflexes, and macular changes) but with clinical variability. Six patients (6%) had optic nerve head drusen (*CEP290* n = 3, *CRB1* n = 3). Other features included macular scars (*NMNAT1* n = 2, *CRB1* n = 2, *RDH12* n = 1), a peripapillary staphyloma (*CABP4* n = 1, *RDH12* n = 1, *RPE65* n = 1, *SPATA7* n = 1), bull`s eye configuration (*CEP290* n = 2, *RPE65* n = 1), central whitish vascular septa (*CRB1* n = 3), glistening spots (*CEP290* n = 1, *CRB1* n = 1), cobblestones (*CEP290* n = 1, *SPATA7* n = 1), and nevi (*CRX* n = 1, *LCA5* n = 1).

OCT (available or evaluable in about half of the patients (51%, 53/105)) generally showed advanced disease with severe photoreceptor loss. In 60% of cases (32/53), the photoreceptors were destroyed, and the ellipsoid zone (EZ), the transition of the photoreceptor inner/outer segments, was not measurable. In 40% of patients (21/53), photoreceptors were measurable centrally. The average EZ was 1215 µm. A morphologically completely intact photoreceptor layer (and accordingly normal EZ) was only found in *CABP4*-variant associated patients with the clinical diagnosis of CSNB (4%, 2/53).

Central foveal thickness (CFT) was reduced to 160 µm on average (Table 1). Exceptions with normal CFT were the genotype groups *CABP4*, *IFT140*, *IQCB1*, and *RPGRIP1* (Table 1). The fovea was particularly atrophic in *AIPL1* (102 ± 34 µm), *CRB1* (146 ± 60 µm), *LRAT* (145 ± 4 µm), *RDH12* (138 ± 72 µm), *RPE65* (137 ± 47 µm), and *TULP1* (131 µm) cases. The outer nuclear layer (ONL; measurable at 27/53) was reduced to 92 µm on average. In addition, three patients had cystoid macular edema (gene groups *CRB1, AIP1*, and *RDH12*). Seven patients had epiretinal gliosis (gene group *CRB1* n = 3, *CEP290* n = 2, and *RDH12* n = 2).

Concomitant diseases were observed in some patients: thyroid disease occurred in the genotype groups *CRB1* (n = 2) and *RDH12* (n = 3), pyelonephritis was seen in a patient with *AIPL1* disease-associated variants, and renal transplantation due to renal failure was necessary for one patient with *IQCB1* variants. Other additional findings included polydactyly, asthma and elevated pancreatic enzymes (n = 1 in *CRB1*), hearing loss (n = 1 in *CRB1*), respiratory arrhythmia (n = 1 in *RDH12*), and scoliosis (n = 1 in *TULP1*).

For further detailed description, we selected the four most frequent genetic subgroups accounting for 70% (73/105) of all cases in our cohort. Some of our cases have been published previously [11,34,35,36,37].

#### 2.2.1. *CEP290*

The clinical and genetic findings of *CEP290*-associated retinal dystrophy seen at our clinical site have been previously published by our group [34]. This was one of the largest subgroups within our cohort: in 22 (21%) patients of 19 families (11 male, 11 female; age 4 to 72 years), 14 unique *CEP290* genotypes, and 17 different *CEP290* variants were revealed. By far, the most common variant, detected in 77% of patients (17/22) with *CEP290*-related pathology, was the deep intronic c.2991+1655A>G variant. Compared to previous studies, we found a surprisingly high number of homozygous c.2991+1655A>G cases (32%, 7/22), indicating this *CEP290* genotype is the most important in Germany. Clinically, all patients showed a severe phenotype. The most common clinical diagnosis was LCA or EOSRD (together with 82%), followed by RP (18%). The clinical diagnosis could not be predicted by the underlying genotype, and phenotypic overlaps existed; however, milder phenotypes were rather observed in compound heterozygous patients or in cases without the deep intronic c.2991+1655A>G variant. Examples of this subgroup are shown in Figure 4.

#### 2.2.2. *CRB1*

Within the cohort, 21% of patients (22/105, 14 male, 8 female, age 8 to 64 years) had disease-causing variants in the *CRB1* gene. The most common diagnosis was RP (69%, 15/22), followed by LCA (27%, 6/22) and CRD (4%, 1/22). Within the cohort, a total of 22 different *CRB1* variants were present. The most frequent variant was c.2843G>A;p.(Cys948Tyr), which was found homozygously in five patients, accounting for 22% of *CRB1* cases and leading to a severe phenotype. In general, *CRB1* patients showed a severe disease, but onset, severity, and rate of progression varied between patients. Characteristic findings included macular atrophy, nummular pigmentation, relative para-arteriolar preservation of the retinal pigment epithelium (RPE), and retinal thickening with loss of lamination on OCT scans—in direct contrast to other forms of LCA/EOSRD where progressive retinal thinning is commonplace (Figure 4).

#### 2.2.3. *RPE65*

The third most frequent genetic subgroup contained 15 patients with *RPE65*-associated retinal dystrophy (15/105, 14%, age 9–51 years; 10 female, 5 male). We identified 13 different *RPE65* variants, 10 patients were homozygous, five (compound) heterozygous. The most frequent variant was c.1451G>T;p.(Gly484Val), which was seen homozygously in five patients (33%, 5/15), all members of a consanguineous family (parents and three children, all affected).

We observed a severe disease pattern with LCA/EOSRD phenotype in 67% of the cases (10/15), with markedly reduced BCVA, nystagmus, and night blindness in most cases (12/15, 80%). *RPE65*-deficiency was associated with reduced or absent autofluorescence on FAF imaging, and on OCT imaging the retinal thickness was variable with a tendency to thin with disease duration (Figure 4).

#### 2.2.4. *RDH12*

Fourteen patients (14/105, 13%, age 3–51 years; 6 female, 8 male) had variants in the *RDH12* gene. A total of 12 different genotypes and 12 *RDH12* variants were identified, with c.806_810del;p.(Ala269GlyfsTer2) (29%, 4/14) being the most frequent. Diagnoses were EOSRD/RP (71%, 10/14) and LCA (29%, 4/14), indicating an onset of the disease somewhat later compared to the other genetic subgroups (Figure 2). We observed a typical fundus phenotype with generalized retinal pigment epithelial and retinal atrophy and minimal intraretinal pigmentation in early childhood, with dense intraretinal bone-spicule pigmentation developing over time and early progressive macular atrophy with foveal thinning, as mentioned before (Figure 4).

## 3. Discussion

To establish the diagnosis of LCA, EOSRD/SECORD, or RP in the light of gene therapy, detailed ophthalmic history, imaging studies, electrophysiological examinations, and, most importantly, genetic analysis is needed. Due to the diversity in genetic background, disease course, and variable onset in infancy or childhood, it is challenging but important for ophthalmologists to obtain a definitive diagnosis in preliminary consultations [3,7,30,38].

When talking about LCA, the difficulty of creating a group collective is striking. We designed the study to include patients with a clinical diagnosis of LCA and patients who are homozygous or (compound) heterozygous for variants in known LCA-associated genes. On the one hand, there were 105 patients with biallelic variants in LCA-associated genes. On the other hand, only 56 patients had a clinical diagnosis of LCA. The diagnosis made by the treating physicians was revised in each case.

One problem lies in the clinical and genetic heterogeneity [3,7,9,34,35,36,37,39] and the lack of a clear definition of LCA. A frequently cited classification by Foxman distinguishes into uncomplicated and complicated LCA, and juvenile and early onset RP [8]. The main criteria are onset of disease and presentation of the ERG, but even these suggested criteria are ambiguous: onset in uncomplicated LCA is before 6 months of age, in complicated LCA, it is “difficult to define”; ERG is “extinguished” in uncomplicated LCA, and “extinguished” or with “residual responses” in complicated LCA [8]. Others specify symptom onset during the first year of life as a criterion [9,10] or do not specify exact inclusion or diagnostic criteria. However, the onset of the disease itself is often difficult to determine, and the ERG may also be non-reproducible (during progression) in other inherited retinal diseases (IRDs)—which was also evident in our cohort.

Recently, a separate group of milder disease phenotypes, the so-called “early-onset severe retinal dystrophy” (EOSRD) or “severe early childhood onset retinal dystrophy” (SECORD), has been described. LCA and EOSRD/SECORD together are the most severe and earliest forms of all IRDs, but there is a fine overlap between the clinical diagnoses of LCA/EOSRD/SECORD/RP. Making a clinical diagnosis remains difficult. Our results underscore that it is useful to consider these as a continuum of phenotypes [29].

Foxman’s classification dates from 1985 and does not consider the genetic background. The first LCA-associated gene, *GUCY2D*, was mapped in 1995 [40]. However, even if the genetics are unambiguous, the clinical diagnosis may still be open because almost all LCA-associated genes queried for this study can also cause other forms of IRD [12]. Given the clinical and genetic heterogeneity, it can be considered a strength of our study that both clinical diagnosis of LCA and disease-causing genotype in LCA-associated genes were taken into account. Thus, our cohort with 16 gene groups and diagnoses of LCA (53%), RP (40%), CRD (5%), and CSNB (2%) covers the entire possible spectrum. Still, several LCA-associated genes were not represented in our cohort, likely due to demographic endowment and extremely rare occurrences.

Of interest, we reported a high rate of variants of uncertain significance (VUS, see Appendix A). If a variant is classified as VUS on the basis of recommended classification principles (e.g., the ACMG guidelines), this is unsatisfactory for physicians, geneticists, and patients alike. However, functional analyses that allow variants to be upgraded to “likely pathogenic” or “pathogenic” are often not possible, especially for missense mutations. Further development of bioinformatic predictions is needed here.

Our study shows that *CEP290, RPE65*, *CRB1*, and *RDH12* are the most important LCA-associated genes in Germany. Their prevalence was 21% and 28% (*CEP290*), 21% and 11% (*CRB1*), 14%, and 23% (*RPE65)*, and 13% and 8% (*RDH12*) for the total cohort and within LCA cases, respectively. To set these numbers into perspective is difficult, as large cohort studies on LCA are missing. Most studies either provide mutation spectra for all IRDs tested in a given setup or for a single gene tested in a certain clinical diagnosis. Yet, Kumaran and coworkers [2] have extensively summarized the clinical and genetic characteristics of LCA/EOSRD and the differential diagnoses to be considered. They stated that the identified LCA-associated genes account for approximately 70–80% of LCA/EOSRD cases, with *GUCY2D*, *CEP290*, *CRB1*, *RDH12*, and *RPE65* being the most common in general. A recent study in 27 Polish families (31 patients) [41] identified causative variants in 37% of cases in *CEP290*, 9 of 10 cases carrying the intronic variant c.2991+1655A>G, 22% in *CRB1*, 11% in *GUCY2D* and *NMNAT1*, two cases in *RPGRIP1*, and single cases with disease-causing variants in *LRAT*, *LCA5*, and *CRX*. In an Italian study on 24 LCA patients [42], the most common genes were found to be *CRB1* (7%), *CEP290*, *IQCB1*, and *GUCY2D* (each 5%), while only single families with variants in *NMNAT1*, *TULP1*, *AIPL1*, *SPATA7*, and *RPGRIP1* were identified. In a Brazilian LCA/EORD cohort of 137 molecularly diagnosed families (152 affected patients) [43], the most commonly mutated genes were *CEP290* (21%), *RPE65* (16%), *CRB1* (14%), *RPGRIP1* (10%), *GUCY2D* (8%), and *RDH12* (8%). A New Zealand study on childhood-onset IRD [44] identified variants most commonly in *ABCA4* and *RS1*, but also in LCA-related genes (*CRB1* 7%, *RPE65* 3%, and two or single cases in *SPATA7*, *CEP290*, *TULP1*, *NMNAT1*, *LCA5*, *GUCY2D*, *CRX*, *RD3*, and *RPGRIP1*). In a recent Chinese study [45] with 37 LCA patients, the three most frequently mutated genes were *CRB1* (27%), *RDH12* (24%), and *RPGRIP1* (19%). Another Chinese study [46] on LCA and EOSRD cases lists as the first five most frequently mutated genes *AIPL1* (11%), *RPGRIP1* (9%), and *CEP290*, *GUCY2D*, and *RPE65* (each 8%) in the LCA patient group, and *RPGR* (12%), *CRB1*, and *RPE65* (each 11%), *RDH12* (7%), and *RP2* (5%) in the EOSRD patient groups. A Japanese study indicates that the most frequently mutated genes were *CRB1*, *NMNAT1*, and *RPGRIP1* [47].

The prevalence of LCA- and RP-associated genes and disease-causing variants is particularly relevant for future gene therapies. The first ophthalmological disease to become the focus of gene therapy was indeed LCA, caused by disease-associated variants in *RPE65* [31,32,48,49,50]. The introduction of wild-type *RPE65* complementary DNA (cDNA) into RPE target cells in animal models and subsequently in humans via viral vector-mediated gene supplementation therapy has resulted in significant improvements in photosensitivity, visual field, and visual function, as demonstrated in several phase II and III clinical studies [48,49,50,51,52,53,54,55]. In an important milestone, voretigen neparvovec (Luxturna^®^, Novartis) was officially approved by the Food and Drug Administration (FDA) in the US in 2017 and by the European Medicines Agency (EMA) in Europe in 2018 for the treatment of IRDs caused by a biallelic disease-causing variants in *RPE65* [31,32]. The success of *RPE65* gene therapy has paved the way for more than 30 gene replacement trials worldwide. Therapeutic clinical trials are ongoing for several genotypes of different IRDs, and hold promise for a cure for a large number of patients in the future [2,3,56,57,58]. Therefore, genetic testing, demographic evaluation of the distribution of different genotypes, and extensive natural history studies in order to describe the course of the disease, identify treatable cells in the retina, define the window of opportunity, and the develop suitable clinical endpoints are highly needed.

In summary, our study fills the gap in the literature on the prevalence of LCA subtypes in Germany and provides for the first time a detailed clinical and genetic characterization of one of the largest cohorts in Europe. The results will help to prepare upcoming gene therapeutic trials and to better counsel affected patients.

## 4. Materials and Methods

The study was conducted at the Institute for Ophthalmic Research, Centre for Ophthalmology, University of Tübingen, Tübingen, Germany. It was approved by the local ethics committee, and informed consent was obtained from all patients. Examinations were performed in accordance with the Code of Ethics of the World Medical Association (Declaration of Helsinki).

### 4.1. Study Design and Study Population

Local databases were queried for (1) patients with a clinical diagnosis of LCA and (2) patients with already established likely biallelic disease-causing variants in LCA-associated genes, despite the clinical diagnosis. Individuals who have been clinically diagnosed with LCA but whose genetic status was unknown were invited to undergo molecular genetic testing.

Clinical data was obtained mainly retrospectively, but in older cases where clinical information was missing, follow-up visits were initiated. Specifically, *CEP290*-related cases were invited for a revisit with respect to a previously published study [34]. All cases were critically re-evaluated, and the correct clinical diagnoses were ascertained on the basis of available genetic and clinical data.

### 4.2. Inclusion Criteria

Male and female patients independent of age with the clinical diagnosis of LCA and patients who were homozygous or carried two heterozygous variants in at least one of the following LCA-associated genes were included: *AIPL1* (Gene ID: 23746; OMIM 604392), *CABP4* (Gene ID 57010: OMIM 608965), *CEP290* (Gene ID: 80184; OMIM 610142), *CRB1* (Gene ID: 23418; OMIM 604210), *CRX* (Gene ID: 1406; OMIM 602225), *GUCY2D* (Gene ID: 3000; OMIM 600179), *IFT140* (Gene ID: 9742; OMIM 614620), *IQCB1* (Gene ID: 9657; OMIM 609237), *KCNJ13* (Gene ID: 3769; OMIM 603208), *LCA5* (Gene ID: 167691; OMIM 611408), *LRAT* (Gene ID: 9227; OMIM 604863), *NMNAT1* (Gene ID: 64802; OMIM 608700), *RD3* (Gene ID: 343035; OMIM 180040), *RDH12* (Gene ID: 145226; OMIM 608830), *RPE65* (Gene ID: 6121; OMIM 180069), *RPGRIP1* (Gene ID: 57096; OMIM 605446), *SPATA7* (Gene ID: 55812; OMIM 609868), and *TULP1* (Gene ID: 7287; OMIM 602280).

### 4.3. Clinical Assessment

Medical history was obtained, and a complete ophthalmological examination was performed—upon availability and feasibility—including psychophysical tests (best corrected visual acuity (BCVA), visual field (VF)), electrophysiology (fullfield and multifocal electroretinography (ERG)), and imaging (fundus photography, fundus autofluorescence (FAF), and spectral-domain optical coherence tomography (SD-OCT)).

Disease onset was evaluated and categorized according to the National Institute of Child Health and Human Development (NICDH) Pediatric Terminology [59] as follows:
-1= Birth; -2= Infancy (beginning 28 days to 12 months of age); -3= Toddler (beginning 13 months -2 years of age); -4= Early childhood (beginning 2 years to 5 years of age); -5= Middle childhood (beginning 6 years to 11 years); -6= Early adolescent (beginning 12 years to 18 years).

Disease onset and clinical symptoms determined the final clinical diagnosis. For easier data analysis, we decided to split the cohort into two subgroups, LCA and RP, respectively. LCA as a diagnosis was given for patients with disease onset in the first 2 years of life, RP was set as a diagnosis for patients with disease onset after the age of 2 years. Further IRD diagnoses were also determined on the basis of clinical findings, if reasonable.

BCVA was measured using a Snellen chart. In patients with severely reduced visual acuity, the ability to count fingers or to perceive hand movements or light, respectively, was tested. Data was converted into Snellen as follows [60,61]:-Counting fingers (CF) = 0.014;-Hand motion (HM) = 0.005;-Light perception (LP), no light perception (NLP) = 0.

For visual field testing, an Octopus 900 perimeter (Haag-Streit International, Köniz, Switzerland) was used. Semiautomated kinetic perimetry was performed using the Goldmann stimuli V4e and III4e, if possible. The size of the fields with stimulus III4e was measured in deg^2^ for the right and left eye, respectively. Fullfield and—if possible—multifocal ERG was performed in patients in order to assure clinical diagnosis. All electrophysiological recordings were performed according to the International Society for Clinical Electrophysiology of Vision Standards (ISCEV) with the E3 Espion system (Diagnosys LLC, Lowell, MA, USA) [62,63]. For FAF and SD-OCT imaging, an SD-OCT (Heidelberg Engineering GmbH, Heidelberg, Germany) was used. If the quality of the OCT images allowed, central foveal thickness (CFT) was measured, and the morphology of the outer nuclear layers were evaluated.

### 4.4. Genetic Analysis

Genetic testing was performed in most cases with comprehensive next generation sequencing gene panels, either in a diagnostic genetic or research context. Methodological details have been described previously [11,33]. In individual cases, genetic testing was performed by conventional Sanger sequencing of individual genes. The variant nomenclature in this manuscript is in accordance with Human Genome Variation Society recommendations [64]. Variant classification in this manuscript was performed using the classification tool from Franklin [65] based on ACMG/AMP guidelines [66] using default settings.

### 4.5. Statistical Analysis

Descriptive statistical analysis was done using JMP (John’s Macintosh Project by SAS Institute, Cary, NC, USA), providing median (m), mean (M), and standard deviation (SD) and spread, respectively. Data for the right and left eye were analyzed separately. The normality of data was assessed by evaluation of the histogram plots, correlation parameters were calculated using Pearson or Spearman analysis, where appropriate.

## Figures and Tables

**Figure 1 ijms-24-08915-f001:**
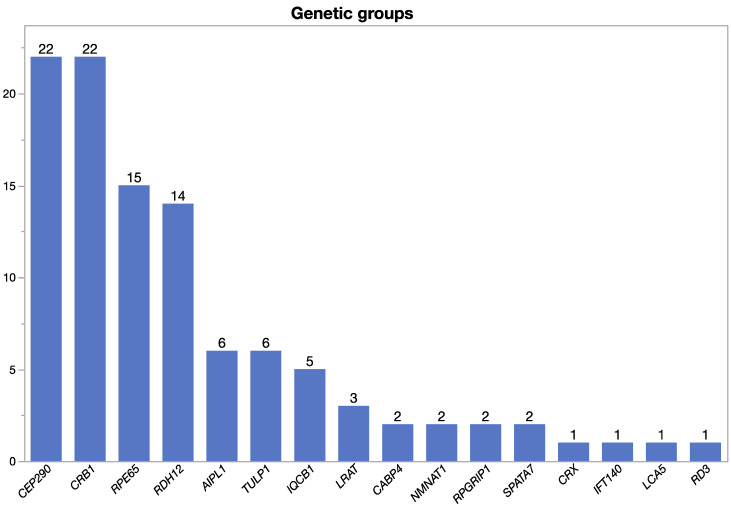
LCA gene distribution within the cohort.

**Figure 2 ijms-24-08915-f002:**
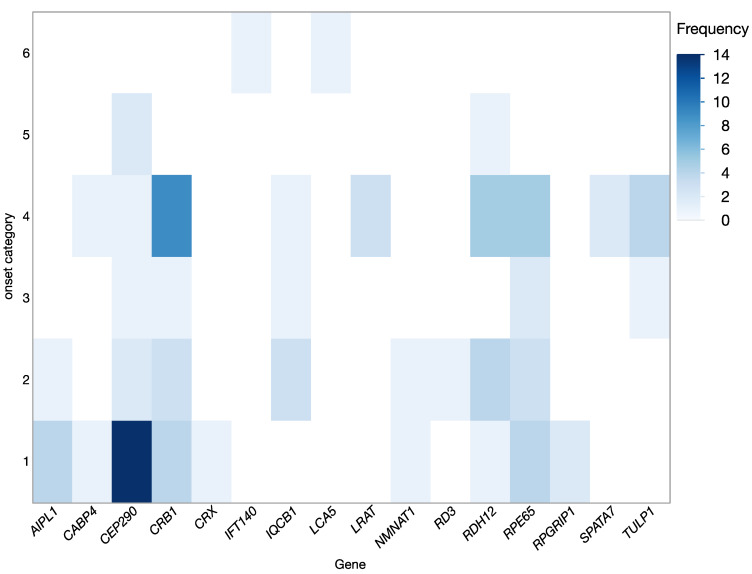
Disease onset in the different genetic subgroups. Disease onset was categorized according to the NICDH Pediatric Terminology as follows: 1 = birth, 2 = infancy (beginning 28 days to 12 months of age), 3 = toddler (beginning 13 months −2 years of age), 4 = early childhood (beginning 2 years to 5 years of age), 5 = middle childhood (beginning 6 years to 11 years), and 6 = early adolescent (beginning 12 years to 18 years), category provided on the Y-axis; genetic subgroup on the X-axis. Frequency, as the number of patients in each genetic group is shown according to the color coding.

**Figure 3 ijms-24-08915-f003:**
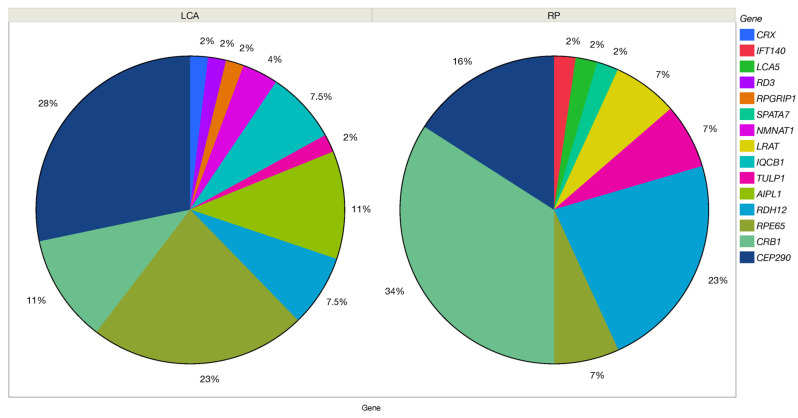
Genetic composition of clinically diagnosed Leber congenital amaurosis (LCA) and Retinitis pigmentosa (RP) cohorts.

**Figure 4 ijms-24-08915-f004:**
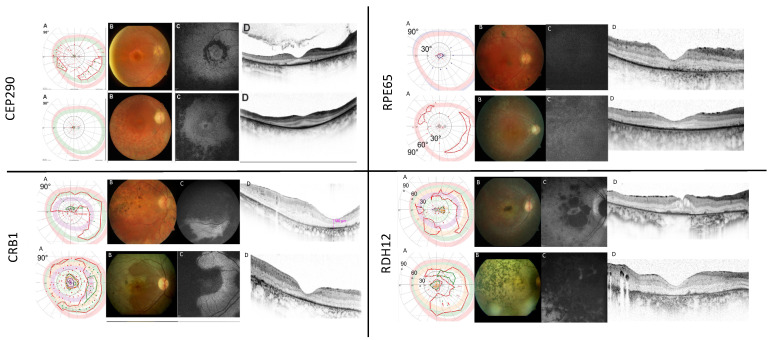
Characteristic clinical findings in the most frequent genetic subgroups (*CEP290*, *CRB1*, *RPE65*, and *RDH12*). (**A**) Kinetic visual fields, tested with V4e (blue), III4e (red), I4e (green), I3e (orange) or I2e (lila) targets, where possible; (**B**) fundus morphology, (**C**) corresponding fundus autofluorescence images, and (**D**) OCT scans. Notice the well-preserved outer nuclear layer on OCT scans in *CEP290* patients (upper left panel), the relative para-arteriolar preservation of the RPE and retinal thickening with loss of lamination on OCT scans in *CRB1* patients (lower left panel), the missing autofluorescence in *RPE65* patients (upper right panel) and the dense pigmentation and macular atrophy in *RDH12* cases (lower right panel).

**Table 1 ijms-24-08915-t001:** Overview of functional and morphological findings of the genetic subgroups and the general cohort. Shown are the number of patients (N), age at data collection, median (m), and mean ± standard deviation (M ± SD) for best corrected visual acuity (BCVA), area of visual field (VF) measured with target III4e and central foveal thickness (CFT) on optical coherence tomography (OCT) images. For N = 1, M ± SD could not be assessed (*). Right eye (OD) and left eye (OS) are depicted separately. Due to nystagmus and hence poor quality of OCT images, CFT could not be assessed in all patients.

Gene	N	Age (Years) M; Range	Eye	BCVA (Snellen)		VF III4e (deg^2^)		OCT CFT (µm)	
				N	M ± SD	N	M ± SD	N	M ± SD
** *AIPL1* **	6	31; 10–49	OD	6	0.044 ± 0.092	2	5325 ± 5140	2	94 ± 0.7
			OS		0.071 ± 0.146		3482 ± 3444		110 ± 57
** *CABP4* **	2	28; 27–28	OD	2	0.365 ± 0.375	2	8847 ± 5048	2	204 ± 11
			OS		0.365 ± 0.375		10370 ± 3896		214 ± 47
** *CEP290* **	22	31; 4–72	OD	22	0.132 ± 0.246	14	804 ± 1741	8	184 ± 41
			OS		0.184 ± 0.294		655 ± 1532		185 ± 37
** *CRB1* **	22	32; 8–64	OD	22	0.101 ± 0.145	14	1476 ± 2474	13	151 ± 59
			OS		0.113 ± 0.174		1250 ± 2437	9	161 ± 64
** *CRX* **	1	46	OD	1	*	1	*	0	*
			OS		*		*		*
** *IFT140* **	1	76	OD	1	*	0	*	1	*
			OS		*		*		*
** *IQCB1* **	5	29; 10–61	OD	5	0.234 ± 0.194	5	36 ± 26	4	229 ± 8
			OS		0.204 ± 0.173		31 ± 39	3	234 ± 3
** *LCA5* **	1	64	OD	1	*	1	*	1	*
			OS		*		*		*
** *LRAT* **	3	39; 38–41	OD	3	0.028 ± 0.025	3	*	2	145 ± 4
			OS		0.094 ± 0.136		*		169 ± 88
** *NMNAT1* **	2	33; 10–55	OD	2	0 ± 0	2	*	0	*
			OS		0 ± 0		*		*
** *RD3* **	1	20	OD	1	*	1	*	1	*
			OS		*		*		*
** *RDH12* **	14	24; 3–51	OD	14	0.110 ± 0.183	8	1332 ± 2468	3	138 ± 72
			OS		0.152 ± 0.215		1805 ± 638		145 ± 105
** *RPE65* **	15	30; 9–51	OD	15	0.062 ± 0.084	15	257 ± 505	7	137 ± 47
			OS		0.059 ± 0.100		329 ± 713	6	154 ± 27
** *RPGRIP1* **	2	29; 23–35	OD	2	0.04 ± 0.014	2	3083 ± 4168	2	194 ± 41
			OS		0.045 ± 0.007		3190 ± 4345	1	*
** *SPATA7* **	2	50; 48–51	OD	2	0.163 ± 0.053	2	2137 ± 3019	2	187 ± 47
			OS		0.143 ± 0.025		1646 ± 237		188 ± 63
** *TULP1* **	6	21; 10–39	OD	6	0.25 ± 0.204	5	2469 ± 3521	5	135 ± 19
			OS		0.272 ± 0.188		2278 ± 2449		134 ± 16
**Total**	**105**	**31 ± 17**	**OD**	**108**	**0.124 ± 0.195**	**80**	**1403 ± 2702**	**53**	**165 ± 43**
			**OS**		**0.147 ± 0.223**		**1277 ± 2544**	**49**	**183 ± 37**

**Table 2 ijms-24-08915-t002:** The most frequent alleles within the general cohort.

Gene	cDNA Position	Amino Acid Position	Number of Alleles
** *CEP290* **	c.2991+1655A>G	p.(Cys998Ter)	27
** *AIPL1* **	c.834G>A	p.(Trp278Ter)	10
** *RPE65* **	c.1451G>T	p.(Gly484Val)	10
** *CRB1* **	c.2843G>A	p.(Cys948Tyr)	8
** *IQCB1* **	c.1558C>T	p.(Gln520Ter)	6
** *RDH12* **	c.806_810del	p.(Ala269GlyfsTer2)	6
** *CRB1* **	c.2234C>T	p.(Thr745Met)	5
** *IQCB1* **	c.1558C>T	p.(Gln520Ter)	5

**Table 3 ijms-24-08915-t003:** Clinical diagnoses within the cohort and their causative genes. Causative genes are ordered by frequency. LCA = Leber congenital amaurosis; EOSRD = early onset severe retinal dystrophy; RP = Retinitis pigmentosa; CRD = Cone-rod dystrophy; CSNB = Congenital stationary night blindness; CD = Cone dystrophy.

**Clinical** **Diagnosis**	**N (Patients)**	**Relative** **Frequency**	**Affected Genes**
**LCA/EOSRD**	56	53%	*CEP290, RPE65, CRB1, AIPL1, IQCB1, RDH12, LRAT, NMNAT1, CRX, RD3, RPGRIP1*
**RP**	42	40%	*CRB1, RDH12, CEP290, TULP1, RPE65*, *AIPL1, LCA5, SPATA7, IFT140, LRAT*
**CRD**	5	5%	*TULP1, CRB1, SPATA7, RPGRIP1*
**CSNB**	2	2%	*CABP4*

**Table 4 ijms-24-08915-t004:** Frequency of associated findings within the genetic subgroups and in the general cohort. N = number of patients; - = non-existent or missing data. ***** = Data for color vision was not documented in 24 patients, thereof 14 patients with BCVA of 0 (Snellen). If these patients were added, 88 patients (84%) were affected by color vision disturbances.

Gene	N	Nystagmus	Strabismus	Oculodigital Sign	Cataract	Keratoconus	Optic Nerve Head Drusen	Night Blindness	Photophobia	Color Vision Disturbance
		N								
** *AIPL1* **	6	6	2	-	2	1	-	2	4	4
** *CABP4* **	2	1	2	-	-	-	-	1	1	2
** *CEP290* **	22	17	9	4	11	4	3	13	13	12
** *CRB1* **	22	11	9	-	10	1	3	18	15	18
** *CRX* **	1	-	-	-	-	-	-	-	-	-
** *IFT140* **	1	-	-	-	1	-	-	1	1	-
** *IQCB1* **	5	3	3	-	2	-	-	4	2	4
** *LCA5* **	1	-	-	-	-	-	-	1	1	-
** *LRAT* **	3	3	1	-	1	-	-	1	-	1
** *NMNAT1* **	2	1	-	-	-	-	-	-	1	1
** *RD3* **	1	1	1	-	-	-	-	-	1	1
** *RDH12* **	14	3	10	-	9	-	-	10	7	11
** *RPE65* **	15	12	6	1	7	-	-	12	9	11
** *RPGRIP1* **	2	2	-	-	-	-	-	1	2	2
** *SPATA7* **	2	1	1	-	2	-	-	2	1	2
** *TULP1* **	6	1	-	-	1	-	-	6	3	5
**General** **(N; frequency)**	**105**	**62** **59%**	**47** **44%**	**5** **5%**	**54** **51%**	**6** **6%**	**6** **6%**	**72** **68%**	**63** **60%**	**74 *** **70% ***

## Data Availability

Data sharing is not applicable to this article.

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
