# Peer review of "Genetic and Clinical Profile of Retinopathies Due to Disease-Causing Variants in Leber Congenital Amaurosis (LCA)-Associated Genes in a Large German Cohort"

_ijms, 2023, doi:10.3390/ijms24108915_

Round 1

Reviewer 1 Report

The paper is well-written, and the methodology is clearly described. However, it would be helpful to provide more details on the selection criteria for the study cohort, especially regarding the clinical diagnosis of the patients.

The statistical analyses performed were appropriate, but it would be helpful to include more information on the distribution of the variants in the different LCA-associated genes and the clinical characteristics of patients with different variants.

The authors reported a high rate of variants of uncertain significance (VUS). It would be helpful to discuss the potential impact of VUS on the clinical management of patients and the need for functional studies to determine their pathogenicity.

The authors should consider adding more discussion on the limitations of the study, especially regarding the potential biases in the patient selection process, the lack of functional studies to confirm the pathogenicity of VUS, and the potential impact of environmental factors on the clinical presentation of patients.

It would be useful to provide more details on the genetic counseling and testing provided to the patients and their families, especially regarding the potential implications of the genetic results on their reproductive choices and future health outcomes.

The conclusions drawn from the study are well-supported by the data presented. However, the authors should consider adding more discussion on the potential clinical implications of their findings and the need for more targeted therapies based on the specific genetic variants detected in each patient.

Minor editing of English language required

Reviewer 2 Report

The study by Zobor et al. provides a comprehensive analysis of the genetic and clinical profile of patients with retinopathies associated with disease-causing variants in Leber Congenital Amaurosis (LCA)-associated genes in a large German cohort. The authors successfully present a clear and well-organized study design and methodology that includes a thorough investigation of local databases, diagnostic-genetic testing, and retrospective collection of clinical data.

The findings of the study are presented in a concise and informative manner, with a focus on the genetic spectrum and associated phenotypic features. The results provide valuable insight into the most frequently mutated genes, CEP290 and CRB1, in the German cohort, as well as the range of clinical diagnoses and variable phenotypic presentations seen among the patients. The study's data analysis is well-supported by descriptive statistical data, making the results easy to comprehend.

Overall, this study provides a significant contribution to the knowledge of the genetic and clinical profile of LCA and highlights the importance of genetic testing and accurate diagnosis in the management of retinal dystrophies. The authors' detailed and comprehensive investigation provides important insights that could be crucial for gene therapeutic trials and management of LCA in the future. The study's design, methodology, and findings are robust and clearly presented, and the authors' conclusions are well-supported by the data.

I have only some minor comments and suggestions that I hope will help you improve your manuscript before publication:

·        -A potential improvement for this study could be the inclusion of multicentric data to increase the robustness and generalizability of the findings. As far as I understood, the data collected in this study is from a single center, which may limit the representativeness of the whole German patient population and the generalizability of the results to other populations. By including data from multiple centers, it may be possible to obtain a more diverse and representative sample of patients, which could improve the reliability of the results and increase the potential for wider application of the findings.

·        -Page 4, Result’s section, 7th line: number of patients carrying RPE65 variants is detailed as 16/109, however in the data provided the number is always 15/105. Please, correct this.

·        -I find it personally intriguing that cataracts are a common finding in individuals with LCA or RP. It is possible that this could be related to an increase in ocular oxidative stress levels caused by the disease. Further investigation into the pathophysiology of cataracts in LCA and RP patients could shed light on potential therapeutic targets for managing this complication. What are your thoughts on this hypothesis?
